# Early Changes in Circulating FGF19 and Ang-2 Levels as Possible Predictive Biomarkers of Clinical Response to Lenvatinib Therapy in Hepatocellular Carcinoma

**DOI:** 10.3390/cancers12020293

**Published:** 2020-01-26

**Authors:** Makoto Chuma, Haruki Uojima, Kazushi Numata, Hisashi Hidaka, Hidenori Toyoda, Atsushi Hiraoka, Toshifumi Tada, Shunji Hirose, Masanori Atsukawa, Norio Itokawa, Taeang Arai, Makoto Kako, Takahide Nakazawa, Naohisa Wada, Shuitirou Iwasaki, Yuki Miura, Satoshi Hishiki, Shuhei Nishigori, Manabu Morimoto, Nobuhiro Hattori, Katsuaki Ogushi, Akito Nozaki, Hiroyuki Fukuda, Tatehiro Kagawa, Kojiro Michitaka, Takashi Kumada, Shin Maeda

**Affiliations:** 1Gastroenterological Center, Yokohama City University Medical Center, Yokohama 232-0024, Japan; kz-numa@orange.zero.jp (K.N.); katsuaki.ogushi@gmail.com (K.O.); akino@yokohama-cu.ac.jp (A.N.); fukuhiro@yokohama-cu.ac.jp (H.F.); 2Department of Gastroenterology, Internal Medicine, Kitasato University School of Medicine, Sagamihara 252-0375, Japan; kiruha@kitasato-u.ac.jp (H.U.); hisashi7@kitasato-u.ac.jp (H.H.); tnakazaw@kitasato-u.ac.jp (T.N.); naow0829@kitasato-u.ac.jp (N.W.); shu1rou0612@gmail.com (S.I.); 3Department of Gastroenterology and Hepatology, Ogaki Municipal Hospital, Ogaki 503-8502, Japan; tkumada@he.mirai.ne.jp (H.T.); tadat0627@gmail.com (T.T.); takashi.kumada@gmail.com (T.K.); 4Gastroenterology Center, Ehime Prefectural Central Hospital, Matsuyama 790-0024, Japan; hirage@gmail.com (A.H.); kojiromichitaka@gmail.com (K.M.); 5Division of Gastroenterology and Hepatology, Department of Internal Medicine, Tokai University School of Medicine, Isehara 259-1193, Japan; hs099212@tsc.u-tokai.ac.jp (S.H.); kagawa@tokai.ac.jp (T.K.); 6Division of Gastroenterology and Hepatology, Nippon Medical School, Tokyo 113-8603, Japan; momogachi@yahoo.co.jp (M.A.); taeangpark@yahoo.co.jp (T.A.); 7Division of Gastroenterology, Nippon Medical School Chiba Hokusoh Hospital, Inzai 270-1694, Japan; itokawa@nms.ac.jp; 8Division of Gastroenterology, Nippon Medical School Musashi Kosugi Hospital, Kawasaki 211-8533, Japan; 9Department of Gastroenterology, Shonan Kamakura General Hospital, Kamakura 247-8533, Japan; kako1210@muf.biglobe.ne.jp; 10Gastroenterology Division, Hadano Red Cross Hospital, Hadano 257-0017, Japan; y.miu.kamodas@gmail.com; 11Division of Gastroenterology, Saiseikai Yokohamashi-Nanbu Hospital, Yokohama 234-0054, Japan; hishikis@nanbu.saiseikai.or.jp; 12Department of Gastroenterology, Yokohama Minami Kyosai Hospital, Yokohama 236-0037, Japan; gorishuu@yahoo.co.jp; 13Hepatobiliary and Pancreatic Medical Oncology, Kanagawa Cancer Center Hospital, Yokohama 241-8585, Japan; m-morimoto@kcch.jp; 14Division of Gastroenterology and Hepatology, Department of Internal Medicine, St. Marianna University School of Medicine, Kawasaki 216-8511, Japan; n2hattori@marianna-u.ac.jp; 15Department of Gastroenterology, Yokohama City University Hospital, Yokohama 236-0004, Japan; smaeda@med.yokohama-cu.ac.jp

**Keywords:** hepatocellular carcinoma, lenvatinib, vascular endothelial growth factor, fibroblast growth factor 19, angiopoietin-2, biomarker, tyrosine kinase inhibitor

## Abstract

Predictive biomarkers of the response of hepatocellular carcinoma (HCC) to Lenvatinib therapy have not yet been clarified. The aim of this study was to identify clinically significant biomarkers of response to Lenvatinib therapy, to target strategies against HCC. Levels of circulating angiogenic factors (CAFs) were analyzed in blood samples collected at baseline and after introducing lenvatinib, from 74 Child-Pugh class A HCC patients who received lenvatinib. As CAF biomarkers, serum vascular endothelial growth factor (VEGF), fibroblast growth factor 19 (FGF19), FGF23, and angiopoietin-2 (Ang-2) were measured using enzyme-linked immunosorbent assays. Results: Significantly increased FGF19 (FGF19-i) levels and decreased Ang-2 (Ang-2-d) levels were seen in Lenvatinib responders as compared to non-responders (ratio of FGF19 level at 4 weeks/baseline in responders vs. non-responders: 2.09 vs. 1.32, respectively, *p* = 0.0004; ratio of Ang-2 level at four weeks/baseline: 0.584 vs. 0.810, respectively, *p* = 0.0002). Changes in FGF23 and VEGF levels at four weeks versus baseline, however, were not significantly different in responders versus non-responders. In multivariate analysis, the combination of serum FGF19-i and Ang-2-d was the most independent predictive factor for Lenvatinib response (Odds ratio, 9.143; *p* = 0.0012). Furthermore, this combination biomarker showed the greatest independent association with progression-free survival (Hazard ratio, 0.171; *p* = 0.0240). Early changes in circulating FGF19 and Ang-2 levels might be useful for predicting clinical response and progression-free survival in HCC patients on Lenvatinib therapy.

## 1. Introduction

Hepatocellular carcinoma (HCC) is one of the most common malignant tumors and the third leading cause of cancer death worldwide [1]. In particular, advanced HCC is known for its poor prognosis [2].

Lenvatinib has recently become available as a new molecular targeted agent for the first-line treatment of unresectable HCC in Japan, the USA, the EU and Asia. Lenvatinib is a multikinase inhibitor that targets vascular endothelial growth factor (VEGF) receptors 1–3 (VEGFR), fibroblast growth factor (FGF) receptors 1–4 (FGFR), platelet-derived growth factor (PDGF) receptor alpha, the rearranged during transfection (RET) oncogene, and KIT [3,4,5,6]. The REFLECT trial showed the non-inferiority of Lenvatinib compared with Sorafenib in terms of the primary endpoint of overall survival (OS), and statistically significant and clinically meaningful improvement in the secondary endpoints of progression-free survival (PFS), time to progression, and objective response rate (ORR) in unresectable HCC [7]. Furthermore, the therapeutic potential of Lenvatinib for unresectable HCC in clinical practice has been reported in several recent literatures [8,9,10,11].

Cancer biomarkers are widely used for prediction of the natural course, prognosis and treatment response in certain malignancies [12,13]. Since the diagnosis of HCC is usually made without performing a biopsy of the tumor, identification of serum biomarkers for the prediction of Lenvatinib response would be of significant benefit for proper selection of patients for treatment [14]. However, to date, there are no established biomarkers that are predictive of the response to therapy or prognostic for disease progression in HCC patients being treated with Lenvatinib.

Angiogenesis is required for tumor growth, progression and metastasis, making it a logical target for antitumor drug development [15]. Increased VEGF expression is significantly associated with angiogenesis and advanced-stage HCC [16]; therefore, the use of VEGFR signaling pathway inhibitors represents a rational and attractive approach to control HCC. Another molecular driver of tumor growth in the pathogenesis of HCC is FGF/FGFR. The FGF/FGFR pathway is part of an escape mechanism to VEGF-targeted anti-angiogenic therapies [17]. The mammalian FGF family comprises 18 ligands, which exert their actions through four highly conserved transmembrane tyrosine kinase receptors (FGFR1, FGFR2, FGFR3, and FGFR4) [18]. Elevated expression of these four receptors have been detected in HCC compared with normal liver tissue [19]. Among them, FGFR4 expression has been observed most frequently in HCC tissues [19]; the ligand of FGFR4 is FGF19. In other studies, elevated serum FGF19 levels were seen in HCC patients [20] and high levels of FGF19 and FGFR4 were associated with poor outcomes in HCC patients [21]. Elevation of FGF23 levels have been shown to be a surrogate marker of FGFR1 inhibition [22], and in particular, circulating FGF23 levels were increased by Lenvatinib therapy in thyroid cancer [23]. Angiopoietin-2 (Ang-2), a relatively novel regulator of angiogenesis that acts through the TEK tyrosine kinase (Tie2) endothelial receptor, has been identified as a potential prognostic biomarker for some types of cancer, including HCC and breast, thyroid and colorectal cancers [14,24,25,26]. Ang-2 can confer compensatory resistance to anti-angiogenesis therapy targeting VEGF [27,28]. Lenvatinib inhibits VEGF- and FGF-driven angiogenesis [29]. Several reports have noted an association between serum Ang-2 expression and clinical outcomes in patients receiving tyrosine kinase inhibitor (TKI) treatments [24,25,26].

Therefore, in this study, four circulating angiogenic factors (CAFs), VEGF, FGF19, FGF23 and Ang-2, were chosen for further investigation of their suitability as predictive biomarkers of Lenvatinib treatment in patients with HCC. This exploratory study aimed (i) to develop a relevant novel predictive model using a serum biomarker for the prediction of response to Lenvatinib, and (ii) to determine its role in predicting PFS in an in-hospital cohort. Specifically, we chose analytes for serum biomarker analysis on the basis of the molecular targets (or ligands of the targeted receptors) of Lenvatinib, or those related to the outcome and/or pathogenesis of HCC, including VEGF, FGF19, FGF23, and Ang-2.

## 2. Results

### 2.1. Patient Characteristics and Treatment Response

Table 1 summarizes the baseline characteristics of the 74 enrolled patients. Forty-nine patients (66.2%) were male and the median age of all patients was 71 years. Thirty-eight (51.4%) and 36 (48.6%) cases were Barcelona clinic liver cancer (BCLC) stage B and C, respectively. Among them, 20 patients had distant metastatic disease and 26 patients had macroscopic portal vein invasion. Eleven patients (14.9%**)** had a past history of sorafenib treatment, including two who were also treated with regorafenib (2.7%). The enrolled patients were divided according to their objective responses to Lenvatinib, i.e., responders (objective response (OR) group) vs. non-responders (non-OR group).

During the treatment period (median, 157 days; range, 33–474 days), 35 patients were found to have an OR (OR group: complete response, 2 patients; partial response, 33 patients) and 39 patients did not have an OR (non-OR group: stable disease, 27 patients and progressive disease, 12 patients).

### 2.2. Association between CAF Levels at Baseline and Treatment Effects

Median levels of FGF19, FGF 23 and VEGF at baseline were not significantly different between the OR and non-OR groups (FGF19, 267.0 vs. 237.4 pg/mL, *p*=0.235; FGF23, 104.6 vs. 101.9 pg/mL, *p* = 0.852; and VEGF, 258.4 vs. 280.0 pg/mL, *p* = 0.540, respectively) (Figure 1a, Appendix A). On the other hand, median levels of Ang-2 at baseline were different between the two groups (7906 vs. 6809 pg/mL in OR and non-OR groups, *p* = 0.024, Figure 2a), with receiver-operating characteristic (ROC) curve analysis showing an area of 0.642, with 60.0% specificity and 60.0% sensitivity (at the cut-off value of 7432. Figure 2e) in discriminating the OR group from the non-OR group. To further verify whether serum levels of CAFs were associated with Lenvatinib treatment response, we compared the ratio of serum levels of FGF19, FGF23, Ang-2 and VEGF at two, four, and eight weeks versus baseline between the OR and non-OR groups at each time point, the results of which are shown below.

### 2.3. Association between Serum Changes in FGF19 Levels and Lenvatinib Treatment Effect

As seen in Figure 1, serum FGF19 levels increased with Lenvatinib treatment. Although the FGF19 ratio at 2 weeks versus baseline was not significantly different between the OR and non-OR groups following 2 weeks of Lenvatinib treatment (Figure 1b), significantly higher FGF19 ratios at 4 weeks and 8 weeks versus baseline were seen in the OR group compared with the non-OR group (ratio at 4 weeks: 2.09 vs. 1.32, *p* = 0.0004, ratio at 8 weeks: 2.19 vs. 1.40, *p* = 0.0015, in the OR and non-OR groups, respectively.) (Figure 1c,d). Furthermore, to verify the ability of serum FGF19 to early predict Lenvatinib response, ROC curve analysis of the FGF19 ratio at 4 weeks versus baseline was performed. Analysis revealed a ROC curve area of 0.726 at the optimal cut-off value of 1.51 for the FGF19 ratio versus baseline, with 68.6% specificity and sensitivity in discriminating the OR group from the non-OR group (Figure 1e).

### 2.4. Association between Serum Changes in Ang-2 Levels and Lenvatinib Treatment Effect

Figure 2 shows that patients who experienced a greater decrease in Ang-2 levels were observed in the OR group compared with the non-OR group at 2 weeks (Ratio of Ang-2 level at 2 weeks versus baseline: 0.709 vs. 0.893, *p* = 0.0041. Figure 2b), 4 weeks (Ang-2 ratio: 0.584 vs. 0.810, *p* = 0.0002. Figure 2c) and 8 weeks (Ang-2 ratio: 0.500 vs. 0.804, *p* < 0.0001. Figure 2d). ROC curve analysis revealed that at the optimal cut-off value of 0.672 for the Ang-2 ratio, the ROC curve area was 0.720 at 2 weeks, 0.766 at 4 weeks with 69.2% specificity and sensitivity (Figure 2f), and 0.833 at 8 weeks, respectively, for discriminating the OR group from the non-OR group. These results indicate that changes in Ang-2 levels at 2 weeks, 4 weeks and 8 weeks versus baseline levels could be superior to absolute Ang-2 levels at baseline for prediction of OR.

### 2.5. Association between Serum Changes in FGF23 and VEGF Levels and Treatment Effects

As seen in Figure 1, serum FGF23 levels were increased with Lenvatinib treatment. Although the ratio of serum FGF23 levels versus baseline were significantly different in the OR and non-OR groups (1.60 vs. 1.28, *p* = 0.032) following 8 weeks of Lenvatinib treatment, FGF23 ratios at 4 weeks and 2 weeks versus baseline were not significantly different between the two groups. Serum VEGF levels were increased with 2 weeks of Lenvatinib treatment, thereafter decreasing at 4 and 8 weeks in both the OR and non-OR groups. The ratio of VEGF versus baseline levels was not different between the OR and non-OR groups at each measurement time point (Appendix A).

Based on these results, from the viewpoint of early prediction of Lenvatinib treatment response, we further investigated the association between changes in serum levels of FGF 19 and Ang-2 at 4 weeks and treatment response in terms of PFS in HCC patients on Lenvatinib treatment.

### 2.6. Relationship between the Combination of Changes in FGF19 and Ang-2 Levels Versus Baseline and Their Predictive Value of Response to Lenvatinib Treatment

We analyzed the correlation between changes in serum levels of FGF 19 and Ang-2 with Lenvatinib treatment. As seen in Figure 3a, there was no strong correlation between the ratios of FGF 19 and Ang-2 at 4 weeks versus baseline (R^2^ = 0.0588); hence, we investigated whether the combination of changes in serum FGF 19 and Ang-2 was able to predict Lenvatinib response more accurately than a change in FGF 19 or Ang-2 alone. Based on the cut-off values of the FGF19 and Ang-2 ratios versus baseline of 1.51 and 0.67, which were determined as being the optimal cut-off values for discriminating the OR from the non-OR group with Lenvatinib treatment, we examined the positive predictive value (PPV) and negative predictive value (NPV) of response to Lenvatinib treatment using these thresholds. We determined that a FGF19 ratio at 4 weeks compared to baseline of > 1.51 represented an increase in FGF19 (FGF19-i), and an Ang-2 ratio of < 0.67 at 4 weeks compared to baseline represented a decrease in Ang-2 (Ang-2-d) (Figure 3b). Figure 3b shows that the PPV of Lenvatinib response using the combination of FGF 19-i and Ang-2-d (80%) was higher than that of FGF 19-i alone (64.9%) and Ang-2-d alone (67.6%). Similarly, the NPV of Lenvatinib response using the combination of FGF 19-i and Ang-2-d (82.7%) was higher than that of both FGF 19-i alone (70.3%) and Ang-2-d alone (70.0%). However, these differences were not statistically significant. 

### 2.7. Factors Associated with Lenvatinib Response

To evaluate the factors affecting Lenvatinib response, the variables of interest (Table 2) were included in uni- and multivariate analyses. For multivariate analysis, variables that had already been identified as risk factors for Lenvatinib response in univariate analysis were selected. Furthermore, the following four covariates related to objective response were taken into account: age, sex, albumin-bilirubin (ALBI) grade and Barcelona clinic liver cancer (BCLC) stage. In multivariate analysis, relative dose intensity (RDI) (Odds ratio (OR), 0.280; 95% confidence interval (CI), 0.092–0.855; *p* = 0.0254) and the combination of serum FGF19-i and Ang-2-d (OR, 9.143; 95% CI, 2.400–34.832; *p* = 0.0012) were independent risk factors for Lenvatinib response (we excluded FGF19-i alone and Ang-2-d alone to avoid confounding).

### 2.8. Association between Changes in FGF19 and Ang-2 Levels and Progression-Free Survival

We performed Kaplan-Meier analysis to determine whether serum changes in FGF19 and Ang-2 expression versus baseline are associated with PFS in patients with HCC receiving Lenvatinib treatment. PFS rates in FGF19-i patients (1-year PFS, 52.8%) were significantly higher than in non-FGF19-i patients (median PFS (mPFS) 151 days, *p* = 0.0111, Figure 4a). Also, PFS rates in Ang-2-d patients (1-year PFS, 56.7%) were significantly higher than in non-Ang-2-d patients (mPFS 138 days, *p* = 0.0011, Figure 4b). Moreover, the duration of PFS was significantly longer in patients with both FGF19-i and Ang-2-d (one-year PFS, 65.6%) than in either FGF19-i alone or Ang-2-d alone patients (mPFS 256 days), and in those with neither FGF19-i nor Ang-2-d (mPFS 124 days *p* < 0.001; Figure 4c). Further, PFS in terms of Ang-2 levels at baseline was not different between the two groups (Figure 4d).

### 2.9. Analysis of Progression-Free Survival Factors among Patients Receiving Lenvatinib Treatment

We next performed comparisons of the PFS rate with regard to various clinical factors. Table 3 shows the results of uni- and multivariate analyses of factors potentially related to PFS in HCC patients under Lenvatinib treatment. For multivariate analysis, variables that had already been identified as risk factors for PFS after Lenvatinib treatment in univariate analysis were selected, and the following two covariates of HCC were taken into account: age and sex. In multivariate analysis, the combination of FGF19-i and Ang-2-d (Hazzard Ratio, 0.171; 95% CI, 0.037–0.793; *p* = 0.0240) was independently associated with poor PFS.

### 2.10. Case Presentations

Three cases of HCC with different changes in FGF19 and Ang-2 levels are shown in Figure 5. A 66-year-old woman presented with massive, advanced HCC predominantly located in the right lobe of the liver, invading the major branch of the portal vein (a). The patient was treated with 8 mg Lenvatinib, which resulted in serum FGF19 increasing significantly (ratio: 4.10) and Ang-2 decreasing (ratio 0.37) at 4 weeks, reduction of tumor volume and a partial therapeutic response (PR) according to mRECIST guidelines using CT images after 8 weeks (b). Furthermore, PR continued for 67 weeks (c). A 77-year-old man presented with massive, advanced HCC predominantly located in the right lobe of the liver (d). Treatment with 8 mg Lenvatinib did not lead to an increase in serum FGF19 levels (ratio 1.23) or a decrease in Ang-2 (ratio 1.24) levels at 4 weeks, and repeat CT scans showed marked tumor progression in the liver (progressive disease; PD) at 7 weeks (e). A 71-year-old woman presented with advanced HCC with a peritoneal metastatic tumor (arrow) (f). Treatment with 8 mg Lenvatinib resulted in a decrease in serum Ang-2 levels at 4 weeks (ratio 0.53), although FGF19 levels did not increase (ratio 1.36). The peritoneal tumor was reduced (arrow) and PR was observed on CT after 8 weeks (g), although repeat CT scans at 37 weeks indicated marked tumor progression (arrow) according to mRECIST (h).

## 3. Discussion

Identification of predictive and prognostic biomarkers, as well as the mechanisms of the effect of Lenvatinib therapy, will help not only in selecting patients who might benefit from treatment, but also in finding combination approaches that offer hope for improved patient outcomes. 

We showed that both increases in serum FGF19 levels and decreases in serum Ang-2 levels early during treatment were associated with a response in patients receiving Lenvatinib and with PFS. In this study, the FGF19 ratio at two weeks/baseline was not significantly different between OR and non-OR groups. Although FGF19 and Ang-2 ratios at four and eight weeks/baseline were significantly different between the OR and non-OR groups, from the viewpoint of early prediction of Lenvatinib treatment response, we have showed that FGF19 and Ang-2 ratio level at 4 weeks/baseline is not only early indicator for Lenvatinib treatment response but also predictor for progression free survival. The current observations suggest that early changes in serum FGF19 and Ang-2 and their combination can potentially be used as prognostic and predictive biomarkers for Lenvatinib therapy.

Ang-2 is known to be produced by cancer cells and plays important roles in regulating tumor angiogenesis [27,28,30]; additionally, it has been reported as a potential prognostic biomarker for certain types of cancers, including HCC [14,24,25,26]. Several studies have reported high circulating (serum or plasma) Ang-2 levels as a predictor of tumor invasiveness or as a diagnostic biomarker of HCC [14,31,32]. Although a direct inhibitory action of Lenvatinib on Ang-2 is uncertain, anti-VEGF is thought to reduce Ang-2 expression in tumor cells by normalizing tumor vessels and making the tumor microenvironment less hypoxic, explaining the decreases in serum Ang-2 levels by Lenvatinib treatment described in several reports [23,25]. Based on these reports and our results, the association between changes in Ang-2 levels and the clinical response to Lenvatinib treatment suggests that antiangiogenic activity contributed to the observed antitumor activity in this study.

FGF/FGFR signaling has been reported to be involved in hepatocarcinogenesis [19,20,21]. Among FGFRs 1-4, the most frequently expressed in clinical HCC samples is FGFR4, the ligand of which is FGF19 [19]. Lenvatinib directly inhibits the activity of FGFRs. Tahara reported that circulating FGF23, the ligand of FGFR1, increased consistently in the Lenvatinib treatment group, suggesting lenvatinib-mediated FGFR inhibition [23]. Our results showed that FGF19 levels increased with Lenvatinib treatment in the OR group compared to the non-OR group. Similar to Tahara’s report, our results supposed a positive feedback mechanism, whereby continuous inhibition of FGFR4 with Lenvatinib treatment resulted in an increase in levels of FGF19, the ligand of FGFR4.

In this study, although FGF23 levels were increased in the OR group as compared to the non-OR group at 8 weeks, the differences were not significant at four weeks. It has been previously reported that, in HCC, increases in FGFR1 (whose ligand is FGF23) levels are not observed as often as increases in levels of FGFR4 (whose ligand is 19). Thus, an increase in serum FGF23 levels with Lenvatinib treatment was not more frequent than that of FGF19 levels.

After initiation of Lenvatinib treatment, serum VEGF levels were increased at two weeks, decreasing thereafter at 4–8 weeks. Our results were similar to Tsuchiya’s report of changes in VEGF with sorafenib treatment [33]. Due to these complex changes in VEGF, prediction of the response to Lenvatinib based on serum levels of VEGF and its ligand VEGFRs might be difficult, although Lenvatinib directly inhibits VEGFRs. Further large-scale analyses are needed to determine whether serum VEGF changes are associated with the response to Lenvatinib treatment.

In addition, we demonstrated that the combination of decrease in serum Ang-2 levels together with an increase in FGF19 levels was more predictive of a response to Lenvatinib and PFS, than a decrease in Ang-2 or increase in FGF19 alone. These results indicated that the antiangiogenic activity of Lenvatinib treatment, indicated by both the decrease in serum Ang-2 and increase in FGF19, contributed to the observed antitumor activity.

In this study, we did not directly compare the usefulness of the biomarkers between patients treated with Lenvatinib and a placebo or other anti-cancer drug. Recently, however, Finn R.S. compared serum biomarkers between patients treated with Lenvatinib and sorafenib; they presented the results of the Lenvatinib arm at the European Society for Medical Oncology (ESMO) [34], which showed decreased Ang-2 levels and increased levels of FGF19 and FGF23 with Lenvatinib therapy. Although our results were similar to Finn et al’s unpublished data, our data showing that the combination of Ang-2 and FGF19 levels is a more sensitive marker for predicting disease progression in HCC patients being treated with Lenvatinib than the Ang-2 or FGF19 level alone is a novel finding. Furthermore, these results might provide clinically important information on molecular targeted therapy for HCC.

This study has some limitations. First, the sample size of patients who received Lenvatinib was relatively small. Evaluation of FGF19 and Ang-2 levels in a large sample is needed to clarify whether the PPV and NPV of the combination of the two markers is statistically significantly higher than that of FGF19 or Ang-2 alone as biomarkers. The statistical outcomes of this analysis also need to be validated in an independent cohort, for which we are currently recruiting patients. Second, observation period was relatively brief (median, 168 days; range, 38–474 days). We performed Kaplan-Meier analysis, which revealed that overall survival (OS) was higher in patients who presented both FGF19-i and Ang-2-d (1-year OS, 85.7%) than in those with either FGF19-i alone or Ang-2-d alone (1-year OS, 62.2%), and in those with neither FGF19-i nor Ang-2-d (median OS, 200 days *p* = 0.0329; Appendix A). Because of the relatively short observation period, we could not comment on whether these markers are involved in overall survival. Furthermore, we did not refer to alternative therapy for non-responders could provide valuable time. A longer observation period is needed to determine whether these markers are predictive of OS and can improve treatment efficacy by suggesting the need for alternative therapy in Lenvatinib non-responders. Future prospective studies are required to address these limitations by recruiting more patients in a multicenter setting using the same protocols.

Currently, several new TKIs are widely utilized in the treatment of HCC, and several clinical trials of the combination of TKIs and immune checkpoint inhibitors are being conducted. Our results have potentially important clinical implications for physicians, enabling them to choose the appropriate treatment strategy for advanced HCC in individual patients.

## 4. Materials and Methods

### 4.1. Ethical Considerations

Written informed consent for obtaining serum and using it in future studies was obtained from all patients. The study protocol conformed to the ethical guidelines of the World Medical Association Declaration of Helsinki and was approved by the ethics committee of the institute (approval number: B180400020).

### 4.2. Study Design and Treatment

In this prospective cohort study, Lenvatinib (Lenvima^®^; Eisai Co., Ltd., Tokyo, Japan) was orally administered to patients with unresectable HCC. Lenvatinib therapy was recommended for patients with unresectable HCC and: (a) the presence of distant metastases; (b) refractory response to previous transcatheter arterial therapies for HCC; or (c) unsuitability for transcatheter arterial therapies due to anatomical reasons. The dose of Lenvatinib was set based on body weight, and it was administered at an initial dose of 12 mg/day for those weighing over 60 kg and 8 mg/day for those weighing less than 60 kg. During the course of treatment, clinicians could adjust the daily dose of Lenvatinib according to the frequency and severity of adverse events (AEs). Lenvatinib was continued until progression of disease (PD) was identified, unmanageable AEs occurred, or if patients wished to discontinue treatment at their own discretion. Inclusion criteria for patients treated with Lenvatinib were: (1) patients with good liver function characterized by Child–Pugh class A; (2) patients whose blood samples were collected prospectively at baseline, and at 2 weeks, 4 weeks, and 8 weeks after commencing therapy; (3) those in whom an image evaluation was performed after Lenvatinib administration for four-eight weeks; (4) Lenvatinib was given for more than 4 weeks; and (5) patients with complete clinical and follow-up data. Between May 2018 and November 2019, we enrolled 101 patients with advanced HCC who were treated with Lenvatinib at our institute or collaborating hospitals in this prospective cohort study. Among them, 27 patients were excluded for the following reasons: 16 patients were Child-Pugh class B, four patients stopped Lenvatinib because of AEs within 4 weeks after starting therapy, four patients were missing serum sample data at 4 weeks after starting lenvatinib, and three patients missed a contrast-enhanced CT scan within 8 weeks. Finally, 74 patients treated with Lenvatinib and whose serum samples were collected at 11 different institutions in Japan (Yokohama City University Medical Center [*n* = 27], Kitasato University Hospital [*n* = 18], Ogaki Municipal Hospital [*n* = 6], Ehime Prefectural Central Hospital [*n* = 6], Tokai University Hospital [*n* = 5], Nippon Medical School Musashi Kosugi Hospital [*n* = 5], Nippon Medical School Chiba Hokusoh Hospital [*n* = 3], Hadano Red Cross Hospital [*n* = 2], Saiseikai Yokohamashi Nanbu Hospital [*n* = 1] and Ashigarakami Hospital [*n* =1]) remained eligible and were enrolled in this study.

### 4.3. Diagnosis and Evaluation of Therapeutic Response

A clinical diagnosis of HCC was made based on the diagnostic criteria of the European Association for the Study of the Liver [35]. [hypervascularity in the arterial phase and washout in the venous or delayed phase by dynamic computed tomography (CT)] Local physicians at each institution evaluated tumors using enhanced CT or MRI at 4–8 weeks after introducing Lenvatinib, in accordance with the modified response evaluation criteria in solid tumors (RECIST) guidelines.^31^ [36]. Objective response (OR) was defined as complete response (CR) plus partial response (PR). Non-OR was defined as stable disease (SD) plus progressive disease (PD). Clinical characteristics, therapeutic response including progression-free survival (PFS; time from the initial administration to progression or any cause of death).

### 4.4. Serum Preparation and Measurement of Cytokine/Angiogenesis Factors

Blood samples from all subjects were centrifuged at 3000× *g* at 4 °C for 10 min, followed by an additional centrifugation at 12,000× *g* for 15 min to completely remove all remaining cells. Serum samples were aliquoted and stored at –80 °C until analysis. VEGF, FGF-19, FGF-23 and Ang-2 levels were measured by enzyme linked immunosorbent assay (ELISA) using the avidin/streptavidin method [25,26]. Absorbance was recorded at 495 nm, and VEGF, FGF-19, FGF-23 and Ang-2 concentrations were read from the standard curve. For pharmacodynamic analysis of changes in serum biomarker levels with treatment, we investigated the levels of VEGF, FGF19, FGF23 and Ang-2 at baseline (74 cases), 2 weeks (67 cases), 4 weeks (74 cases) and 8 weeks (59 cases) after introducing lenvatinib. Furthermore, we compared the ratio of serum levels of FGF19, FGF23, Ang-2 and VEGF at 2, 4 and 8 weeks with those at baseline in Lenvatinib responders and non-responders at each time point.

### 4.5. Statistical Analysis

Data are expressed as the mean ± standard error of the mean (SEM). Significant differences were detected using non-parametric testing. Student’s t-test was performed to compare the differences in serum VEGF, FGF19, FGF23 and Ang-2 levels between responders and non-responders to Lenvatinib treatment. Independent factors associated with response were assessed using logistic regression analysis. The cumulative PFS rate was calculated from the first date of Lenvatinib treatment using the Kaplan-Meier method. Differences were evaluated by log-rank testing. Independent factors for PFS were assessed using the Cox proportional hazard regression model. Statistical analyses were performed using Stat View software (version 5.0; SAS Institute, Cary, NC, USA). Values of *p* < 0.05 were considered significant.

## 5. Conclusions

The analysis of Lenvatinib-induced changes in the serum levels of biomarkers related to angiogenesis suggest that the inhibition of angiogenesis might correlate with clinical outcomes in patients with HCC. Further study of the levels of angiogenic biomarkers and their potential relation to clinical outcomes with Lenvatinib treatment for HCC appears warranted and might impact treatment decision-making.

## Figures and Tables

**Figure 1 cancers-12-00293-f001:**
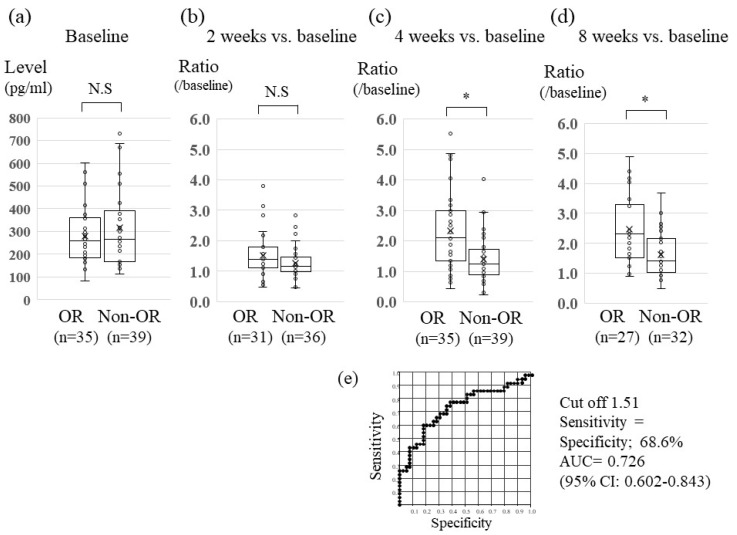
Association between serum FGF 19 levels and lenvatinib treatment response. Distribution of serum fibroblast growth factor (FGF) 19 levels at baseline (**a**), and the ratio versus baseline of FGF19 levels at 2 weeks (**b**), 4 weeks (**c**), and 8 weeks (**d**) between the OR (objective response) and non-OR groups. Data are shown as median values (10th–90th percentile ranges). The Mann-Whitney or Kruskal-Wallis test was used to determine statistical signi ficance. * *p* < 0.05, N.S: non-significant. (**e**) Receiver-operating characteristic curve (ROC) analyses of the ratio of FGF19 at 4 weeks versus baseline for differentiating patients in the OR group versus the non-OR group. AUC, area under the ROC curve.

**Figure 2 cancers-12-00293-f002:**
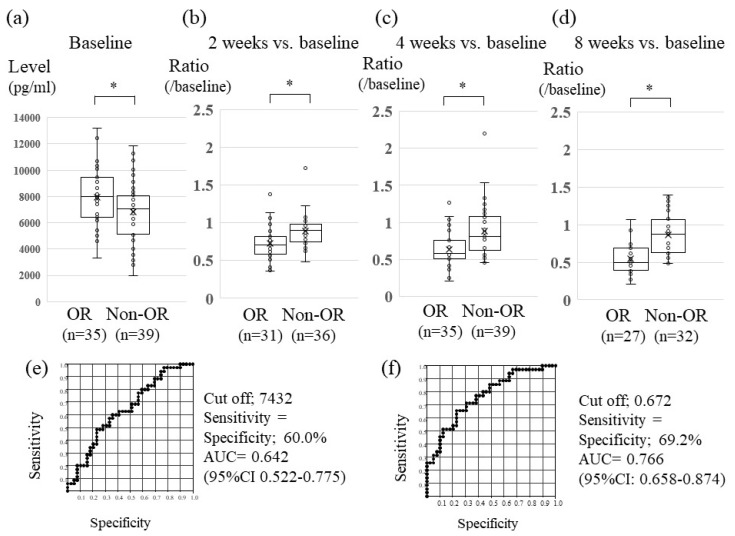
Association between serum Ang-2 level and Lenvatinib treatment response. Distribution of serum angiopoietin-2 (Ang-2) levels at baseline (**a**), and ratio of Ang-2 levels versus baseline at 2 weeks (**b**), 4 weeks (**c**), and 8 weeks (**d**) between the OR (objective response) and non-OR groups. Data are shown as median values (10th–90th percentile ranges). The Mann-Whitney or Kruskal-Wallis test was used to determine statistical significance. * *p* < 0.05, N.S: non-significant. ROC analyses of the level of Ang-2 at baseline (**e**) and the Ang-2 ratio at 4 weeks/baseline (**f**) for differentiating patients in the OR group versus the non-OR group. AUC, area under the ROC curve.

**Figure 3 cancers-12-00293-f003:**
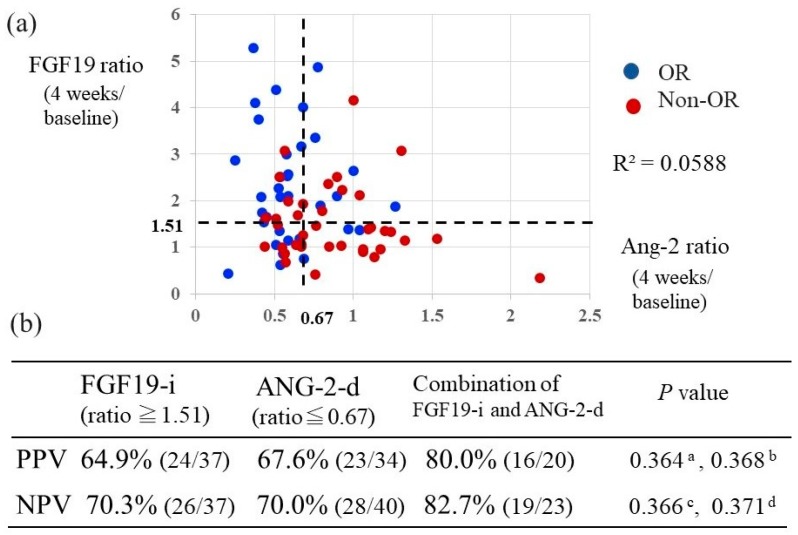
Relationship between changes in FGF19 and Ang-2 levels and predictability of response to Lenvatinib treatment. (**a**) Correlation between the ratio (at 4 weeks/baseline) of serum levels of fibroblast growth factor (FGF) 19 and angiopoietin-2 (Ang-2) with Lenvatinib treatment. R^2^, coefficient of determination (**b**) Positive predictive value (PPV) and negative predictive value (NPV) of Lenvatinib response using the cut-off ratio of fibroblast growth factor (FGF) 19 and angiopoietin-2 (Ang-2) determined in this study. ^a^, PPV of both FGF19-i and Ang-2-d vs. PPV of FGF19-i; ^b^, PPV of both FGF19 and Ang-2 vs. PPV of Ang-2-d; ^c^, NPV of both FGF19-i and Ang-2-d vs. NPV of FGF19-i; ^d^, NPV of both FGF19-i and Ang-2-d vs. NPV of Ang-2-d.

**Figure 4 cancers-12-00293-f004:**
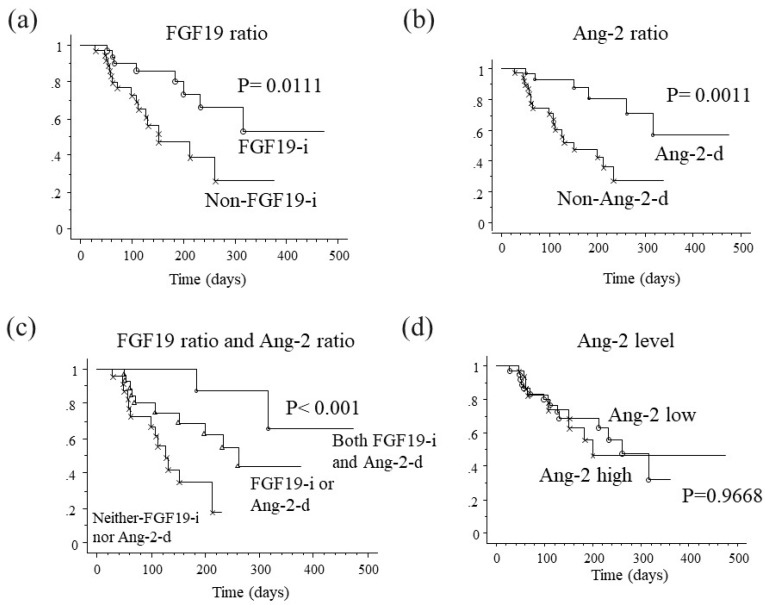
Association between changes in FGF19 and Ang-2 levels and progression-free survival. Kaplan-Meier analysis of 74 hepatocellular carcinoma (HCC) patients who received Lenvatinib treatment, stratified according to ratio of fibroblast growth factor (FGF) 19 (grouped based on a cut-off ratio of 1.51 at 4 weeks versus baseline) (**a**), ratio of angiopoietin-2 (Ang-2) (grouped by a cut-off ratio of 0.67 at 4 weeks versus baseline) (**b**), ratio of FGF19 and Ang-2 (**c**), and level of Ang-2 (**d**). FGF19-i represents a FGF19 ratio at 4 weeks/baseline of >1.51 and Ang-2-d represents an Ang-2 ratio at 4 weeks/baseline of <0.67. Ang-2 high represents an Ang-2 baseline levels >7432 pg/mL, Ang-2 low represents an Ang-2 baseline levels < 7432 pg/mL.

**Figure 5 cancers-12-00293-f005:**
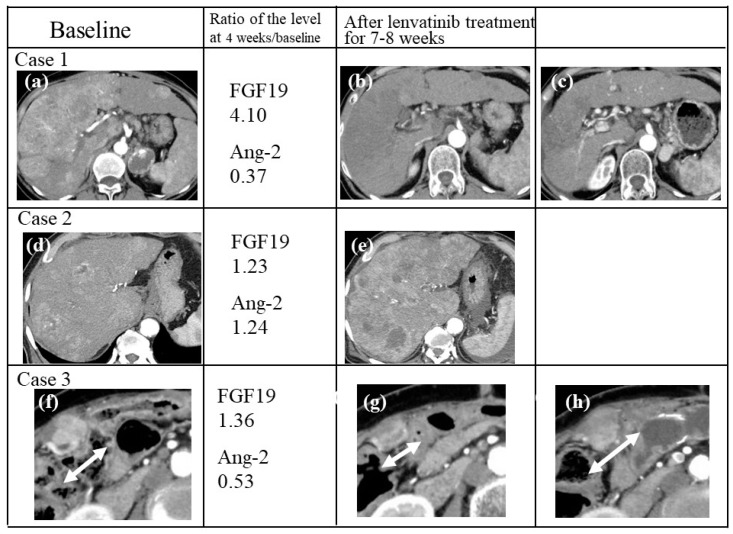
Representative liver lesions of HCC at baseline on arterial phase computed tomography (CT) (**a**, **d**, **f**) and CT scans of each patient after Lenvatinib treatment for 7 weeks (**e**) and 8 weeks (**b**, **g**), and at 67 weeks (**c**) and 37 weeks (**h**).

**Table 1 cancers-12-00293-t001:** Patients characteristics.

Variable	*n* = 74
Median age, years	71 (46–95)
Sex (Male/Female), *n* (%)	49/25: 66.2/33.8%
Cause of HCC (HBV/HCV/NBNC), *n* (%)	12/27/35: 16.2/36.5/47.3%
BMI	22.7 (16.9–35.7)
Child-Pugh score 5/6, *n*: %	52/22: 70.3/29.7%
mALBI grade (1,2a/2b,3)	30/20/23/1: 40.5/27.0/31.1/1.4%
PS (0/1)	67/7: 90.5/9.5%
Extrahepatic metastasis, *n* (%)	20 (27.0%)
MVI, *n* (%)	26 (35.1%)
BCLC (B/C)	38/36: 51.4/48.6%
TNM (II/III/IVA/IVB) LCSGJ 6th	4/36/14/20: 5.4/48.6/18.9/27.0%
TKI 1st line / 2nd line/3rd line~	63/8/3: 85.1/10.8/4.1%
Past history of TACE, *n* (%)	56 (75.7%)
AFP (ng/mL)	38.0 (1.0–262,413)
DCP (AU/mL)	468 (10–290,000)

HBV, hepatitis B virus; HCV, hepatitis C virus; BMI, body mass index; mALBI grade, modified albumin-bilirubin grade; PS, Eastern Cooperative Oncology Group performance status; MVI, major venous invasion; TKI, tyrosine kinase inhibitor; BCLC, Barcelona clinic liver cancer; TACE, transcatheter arterial chemoembolization; TNM stage, tumor node metastasis stage; LCSGJ 6th, the Liver Cancer Study Group of Japan, 6th edition; AFP, alpha-fetoprotein; DCP, des-gamma-carboxy prothrombin. HCC: hepatocellular carcinoma; NBNC: non-hepatitis B virus and non-hepatitis C virus.

**Table 2 cancers-12-00293-t002:** Relationship between clinical factors and levels of biomarkers and objective response to Lenvatinib.

Variable		Univariate Analysis			Multivariate Analysis		
		OR	95% CI	*p* Value	OR	95% CI	*p* Value
Age (years)	<71	1.235	0.495–3.082	0.6505	0.677	0.227–2.017	0.4834
Sex	Male	0.818	0.311–2.155	0.6851	1.071	0.358–3.203	0.9020
HCV status	Positive	1.932	0.733–5.090	0.1828			
Child-Pugh score	5	1.111	0.409–3.021	0.8364			
mALB grade	1+2a	1.618	0.594–4.404	0.3464	1.325	0.418–4.197	0.6320
PS	0	2.426	0.440–13.40	0.3092			
EHM	Absent	1.135	0.405–3.179	0.8097			
MVI	Absent	1.739	0.658–4.598	0.2645			
BCLC stage	B	1.25	0.501–3.120	0.6325	0.754	0.325–2.455	0.5921
TNM stgae LCSGJ 6th	II + III	1.579	0.627–3.974	0.3320			
Past history of TKI	Naïve	1.091	0.302–3.946	0.8945			
Past history of TACE	Naïve	0.889	0.294–2.691	0.8349		
AFP (ng/mL )	<38	0.818	0.327–2.046	0.6678			
DCP (mAU/mL)	<468	2.41	0.947–6.131	0.0649			
RDI	<0.8	0.363	0.141–0.934	0.0357	0.280	0.092–0.855	0.0254
FGF19-i	Positive	4.364	1.644–11.58	0.0031			
Ang-2-d	Positive	4.879	1.819–13.09	0.0016			
Both FGF19–i and Ang-2-d	Positive	7.368	2.154–25.21	0.0015	9.143	2.400–34.832	0.0012
Baseline Ang-2 level	<7432	0.464	0.183-1.175	0.1053			

OR, odds ratio; HCV, hepatitis C virus; mALBI grade, modified albumin-bilirubin grade; TKI, tyrosine kinase inhibitor, PS, Eastern Cooperative Oncology Group performance status; EHM, extrahepatic metastases; MVI, major venous invasion; AFP, alpha-fetoprotein; DCP, des-gamma-carboxy prothrombin; TNM stage, tumor node metastasis stage; LCSGJ 6th, the Liver Cancer Study Group of Japan, 6th edition; BCLC, Barcelona clinic liver cancer; TACE, transcatheter arterial chemoembolization; FGF19, fibroblast growth factor 19; Ang-2, Angiopoietin-2; FGF19-i, FGF19 ratio (4 weeks/baseline) is ≥1.51; Ang-2-d, Ang-2 ratio (4 weeks/baseline) is ≤0.67.

**Table 3 cancers-12-00293-t003:** Association between clinical factors and levels of biomarkers and progression-free survival.

Variable		Univariate Analysis			Multivariate Analysis		
		HR	95% CI	*p* Value	HR	95% CI	*p* Value
Age (years)	<71	1.092	0.492–2.422	0.8295	1.150	0.494–2.674	0.7462
Sex	Male	2.142	0.976–4.701	0.0575	2.041	0.880–4.733	0.0964
HCV status	Positive	0.789	0.357–1.742	0.5573			
Child-Pugh score	5	0.486	0.213–1.112	0.0876			
mALB grade	1 + 2a	0.361	0.159–0.823	0.0153	0.442	0.180–1.089	0.0760
PS	0	0.468	0.173–1.265	0.1344			
EHM	Absent	0.524	0.235–1.170	0.1150			
MVI	Absent	0.323	0.146–0.719	0.0056	0.528	0.167–1.673	0.2778
BCLC stage	B	0.293	0.122–0.706	0.0062	0.888	0.200–2.060	0.4563
TNM stgae LCSGJ 6th	II + III	0.357	0.153–0.832	0.0170			
Past history of TKI	Naïve	1.52	0.450–5.130	0.5010			
Past history of TACE	Naïve	0.776	0.280–2.150	0.6250			
AFP (ng/mL)	<38	0.526	0.237–1.163	0.1125			
DCP (mAU/mL)	<468	0.550	0.243–1.241	0.1500			
RDI	<0.8	1.796	0.805–4.008	0.1796			
FGF19-i	Positive	0.346	0.147–0.814	0.015			
Ang-2-d	Positive	0.235	0.092–0.602	0.0025			
Both FGF19-i and Ang-2-d	Positive	0.237	0.100–0.561	0.0010	0.171	0.037–0.793	0.0240
Baseline Ang-2 level	<7432	0.983	0.444–2.179	0.9668			

HR, Hazard ratio; HCV, hepatitis C virus; mALBI grade, modified albumin-bilirubin grade; TKI, tyrosine kinase inhibitor, PS, Eastern Cooperative Oncology Group performance status; EHM, extrahepatic metastases; MVI, major venous invasion; AFP, alpha-fetoprotein; DCP, des-gamma-carboxy prothrombin; TNM stage, tumor node metastasis stage; LCSGJ 6th, the Liver Cancer Study Group of Japan, 6th edition; BCLC, Barcelona clinic liver cancer; TACE, transcatheter arterial chemoembolization; FGF19, fibroblast growth factor 19; Ang-2, Angiopoietin-2; FGF19-i, FGF19 ratio level (4 weeks/baseline) is ≥1.51; Ang-2-d, Ang-2 ratio level (4 weeks/baseline) is ≤0.67.

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
