# Peer review of "Early Changes in Circulating FGF19 and Ang-2 Levels as Possible Predictive Biomarkers of Clinical Response to Lenvatinib Therapy in Hepatocellular Carcinoma"

_cancers, 2020, doi:10.3390/cancers12020293_

Round 1
Reviewer 1 Report
The aim of this review publication was to “identify clinically significant biomarkers of response to lenvatinib therapy, to target strategies against HCC”. The paper can be interesting also from clinical point of view. According to the authors “to date, there are no established biomarkers that are predictive of the response to therapy or prognostic for disease progression in HCC patients being treated with lenvatinib”. However, the authors cite some unpublished research results of Finn RS et al. from the congress in 2018, which shows similar results, namely decrease Ang-2 levels and increased levels of FGF19 in patients treated with lenvatinib. It is a pity that these results cannot be compared with the results of the reviewed paper, because there is no access to these data. It would be worth adding some details and directions for further research.
I advise to make the careful text corrections that will improve the readability and quality of the work. Especially subchapter „Results” should be improved. The descriptions are not enough professional and misleading. The authors should comment on their ratio of marker levels to baseline marker level in case of all biomarkers in OR and non-OR group. For example I don’t understand the Suppl. Fig. 1b, c, d (FG23 levels) – are these values also ratios? or “real” (true) and median levels?
I recommend to change or standarize everywhere on Figures (1, 2, S2, etc.) the word “baseline” (“base line” or “baseline” in spelling), define the name of “RATIO level” and add “level” on Figures; please compare and check all the descriptions under figures with the description on the Figures. The word „median” is not necessary on the Figures, because these values are well visible. The names of the statistical tests are not important and necessary under Figures. On Fig. 1 c, please correct the value “1.23” (1.32 in text) and Fig. 1 e - specificity is 0.686 or 68.6%?; what is 1-Specificity? (also on Fig. 2f). Please remove the abbreviation LEN in line 179, or use everywhere the abbreviation LEN instead of lenvatinib. On Figure 3 b please add the value P, which shows the significant differences between PPV and NPV for alone biomarkers and combinations of both biomarkers.
I recommend the authors to improve (and a little expand) the header descriptions of all Tables and correct some descriptions under the Tables, e.g. under Table 1 – should be „mALBI grade” – „modified” or „ALBI grade” (line 206)?; Table 2 – description should be „Clinical factors and levels of biomarkers…to objective response to lenvatinib”; Figure 5 - Line 256 – should be 8 weeks (g) and 37 weeks (h) – „h” is missing. The Figure 5 has a little vague description. Please correct it.In general, please clearly define the term "ratio" in the text and use it in a uniform manner in the figures (compare Fig. 1 with “normal” ratio and Fig. 5 a and b with “stratified” ratio). Every description under Figure and the results themselves shown in the figure will improve the readability and quality of the work. In Discussion chapter I am asking for more reliable comment what is the clinical significance of the corrected (baseline/level at 2, -4, 8-week of treatment) of these biomarkers in the early phase of HCC treatment?
Small errors: 1. Line 134, , 151 etc. – “pharmacodynamics is the science of the drug's effect on individual organs and the mechanism of this action”, should be replaced by another term; 2. Line 45 - CAF(s) instead of CAF?; line 95- lenvatinib – without underlining; 3. Table 2 - use “:” or brackets () in some places; standarize the descriptions; 4. Line 166 – “OR group” - without bolding the words; 5. Line 385 – “Objective” response; 6. Line 303 the citation is not correct – “Richard S.F.”; 7. The citations according the Journal should be in in square brackets in all text, please improve it in line 347, 350, 360, etc.
Author Response
We are grateful to the reviewers for the critical comments and useful suggestions that have helped us to improve our paper. As indicated in the responses that follow, we have taken all these comments and suggestions into account in the revised version of our paper.
We added Satoshi Kobayashi and Nobuhiro Hattori as a coauthor, and have revised the Conflicts of Interest statement as follows (Lines 463-465):
“Hidenori Toyoda has received lecture fees from AbbVie, MSD and Bayer Yakuhin Ltd. Atsushi Hiraoka has received lecture fees from Eisai Co., Ltd., Bayer Yakuhin Ltd., and Otsuka Pharmaceutical Co., Ltd.”
The aim of this review publication was to “identify clinically significant biomarkers of response to lenvatinib therapy, to target strategies against HCC”. The paper can be interesting also from clinical point of view. According to the authors “to date, there are no established biomarkers that are predictive of the response to therapy or prognostic for disease progression in HCC patients being treated with lenvatinib”. However, the authors cite some unpublished research results of Finn RS et al. from the congress in 2018, which shows similar results, namely decrease Ang-2 levels and increased levels of FGF19 in patients treated with lenvatinib. It is a pity that these results cannot be compared with the results of the reviewed paper, because there is no access to these data. It would be worth adding some details and directions for further research.
Response
As suggested by the reviewer, we described a comparison between our results and the unpublished research results of Finn RS et al’s study from the congress in 2018, and have accordingly revised the following statements (lines 329-342):
“In this study, we did not directly compare the usefulness of the biomarkers between patients treated with lenvatinib and a placebo or other anti-cancer drug. Recently, however, Finn RS compared serum biomarkers between patients treated with lenvatinib and sorafenib; they presented the results of the lenvatinib arm at the European Society for Medical Oncology(ESMO) [34], which showed decreased Ang-2 levels and increased levels of FGF19 and FGF23 with lenvatinib therapy. Although our results were similar to Finn RS et al’s unpublished data, our data showing that the combination of Ang-2 and FGF19 levels is a more sensitive marker for predicting disease progression in HCC patients being treated with lenvatinib than the Ang-2 or FGF19 level alone is a novel finding. Furthermore, these results might provide clinically important information on molecular targeted therapy for HCC.”
I advise to make the careful text corrections that will improve the readability and quality of the work. Especially subchapter „Results” should be improved. The descriptions are not enough professional and misleading. The authors should comment on their ratio of marker levels to baseline marker level in case of all biomarkers in OR and non-OR group. For example I don’t understand the Suppl. Fig. 1b, c, d (FG23 levels) – are these values also ratios? or “real” (true) and median levels ?
Response
We are sorry for the error in supplementary Figures 1 and 2.
As suggested by the reviewer, we have added the values of “Ratio vs. baseline” in Supplementary Figures 1 and 2.
I recommend to change or standarize everywhere on Figures (1, 2, S2, etc.) the word “baseline” (“base line” or “baseline” in spelling), define the name of “RATIO level” and add “level” on
Response
As suggested by the reviewer, we revised to “baseline” on Figure 1, 2, S1, 2.
I apologize for the vague description. Native speakers of English proofread our manuscript and he suggested to not use the terminology ‘ratio level'. Hence, I have changed the description in the text to “ratio at x weeks versus baseline”.
Figures; please compare and check all the descriptions under figures with the description on the Figures. The word „median” is not necessary on the Figures, because these values are well visible. The names of the statistical tests are not important and necessary under Figures.
Response
As suggested by the reviewer, we have deleted the words “median” and “p value” in Figures 1 and 2 and Supplementary Figures 1 and 2 and revised the description appropriately. Furthermore, we have added the following sentences in the text and phrases in Figures 2 and 3:
“(FGF19, 267.0 vs 2374.4 pg/ml, P=0.235; FGF23, 104.6 vs 101.9 pg/ml, P=0.852; VEGF, 258.4 280.0 pg/ml, P=0.540, respectively)” (lines 128-129)
“*P<0.05, N.S: non-significant.” (Lines 155-156, 175, 439, 443)
On Fig. 1 c, please correct the value “1.23” (1.32 in text) and Fig. 1 e - specificity is 0.686 or 68.6%?; what is 1-Specificity? (also on Fig. 2f). Please remove the abbreviation LEN in line 179, or use everywhere the abbreviation LEN instead of lenvatinib.
Response
We are sorry for these mistakes.
Our results were “1.32”, and accordingly, we deleted “1.23, median value” in Figure 1c.
We have changed “1-Specificity” to “Specificity” and revised Figures 1e, 2e and f, and deleted the abbreviation LEN on line 179.
On Figure 3 b please add the value P, which shows the significant differences between PPV and NPV for alone biomarkers and combinations of both biomarkers.
Response
As suggested by the reviewer, we added P values in Figure 3b. PPV and NPV for the combination of the two markers was higher than that of FGF19 or Ang-2 alone as biomarkers, although the differences were not statistically significant. Based on this result, we have added the following sentences in the text:
“However, these are not statically difference.” (line 208-209)
“ a, PPV of both FGF19-i and Ang-2-d vs. PPV of FGF19-i; b, PPV of both FGF19 and Ang-2 vs. PPV of Ang-2-d; c, NPV of both FGF19-i and Ang-2-d vs. NPV of FGF19-i; d, NPV of both FGF19-i and Ang-2-d vs NPV of Ang-2-d” (line 216-218)
“Evaluation of FGF19 and Ang-2 levels in a large sample is needed to clarify whether the PPV and NPV of the combination of the two markers is statistically significantly higher than that of FGF19 or Ang-2 alone as biomarkers.” (lines 340-342)
I recommend the authors to improve (and a little expand) the header descriptions of all Tables and correct some descriptions under the Tables, e.g. under Table 1 – should be „mALBI grade” – „modified” or „ALBI grade” (line 206)?; Table 2 – description should be „Clinical factors and levels of biomarkers…to objective response to lenvatinib”; Figure 5 - Line 256 – should be 8 weeks (g) and 37 weeks (h) – „h” is missing. The Figure 5 has a little vague description. Please correct it.
Response
As suggested by the reviewer, we have changed the description to “modified ALBI grade” and revised the text as follows:
“Relationship between clinical factors and levels of biomarkers and objective response to lenvatinib” (Line 230)
“Association between clinical factors and levels of biomarkers and progression-free survival” (line 252)
“37 weeks (h)” (Line 280)
In general, please clearly define the term "ratio" in the text and use it in a uniform manner in the figures (compare Fig. 1 with “normal” ratio and Fig. 5 a and b with “stratified” ratio).
Response
At first, I apologize for the vague description. The same ratio has been shown in Figure 1 and Figure 5. As suggested, we defined the term “ratio” in the text, and used it in a uniform manner in the figures and the revised text. We also revised the description as “ratio/baseline” in the text.
Every description under Figure and the results themselves shown in the figure will improve the readability and quality of the work. In Discussion chapter I am asking for more reliable comment what is the clinical significance of the corrected (baseline/level at 2, -4, 8-week of treatment) of these biomarkers in the early phase of HCC treatment ?
Response
In this study, the FGF19 ratio at 2 weeks/baseline was not significantly different between OR and non-OR groups. Although FGF19 and Ang-2 ratios at 4 and 8 weeks/baseline were significantly different between the OR and non-OR groups, from the viewpoint of early prediction of lenvatinib treatment response, there was an association between objective response and progression-free survival with FGF19 and Ang-2 ratios versus baseline at 4 weeks.
As suggested by the reviewer, we have added the following sentences in the discussion (lines 277-279):
“In this study, the FGF19 ratio at 2 weeks/baseline was not significantly different between OR and non-OR groups. Although FGF19 and Ang-2 ratios at 4 and 8 weeks/baseline were significantly different between the OR and non-OR groups, from the viewpoint of early prediction of lenvatinib treatment response, we have showed that FGF19 and Ang-2 ratio level at 4 weeks/baseline is not only early indicator for lenvatinib treatment response but also predictor for progression free survival”
Small errors: 1. Line 134, , 151 etc. – “pharmacodynamics is the science of the drug's effect on individual organs and the mechanism of this action”, should be replaced by another term;
Line 45 - CAF(s) instead of CAF?; line 95- lenvatinib – without underlining; Table 2 - use “:” or brackets () in some places; standarize the descriptions; Line 166 – “OR group” - without bolding the words; Line 385 – “Objective” response; Line 303 the citation is not correct – “Richard S.F.”; The citations according the Journal should be in in square brackets in all text, please improve it in line 347, 350, 360, etc.
Response
As suggested by the reviewer, we have revised the sentences as follows:
“Association between serum changes in FGF19 levels and lenvatinib treatment effect” (line 139)
“Association between serum changes in Ang-2 levels and lenvatinib treatment effect” (line 160)
“circulating angiogenic factors (CAFs)” (line 46)
“lenvatinib” – without underlining. (line 97)
“Objective response” (line 403) “Finn RS” (line 331)
We deleted the brackets () in Table 2 and Table 3 and revised the text accordingly. We revised line 166 (original text)– “OR group” - without bolding the words;
We have revised and changed the reference citations as follows:
“regulating tumor angiogenesis [27, 28, 30]” (line 296)
“as a diagnostic biomarker of HCC [14, 31, 32].” (line 299)
“sorafenib treatment [33].” (line 3320)
“Study of the Liver. [35].” (line 399)
“Response Evaluation Criteria in Solid Tumors (RECIST) guidelines [36].” (line 403)
Reviewer 2 Report
The manuscript by Chuma et al. “Early changes in circulating FGF19 and Ang-2 levels as possible predictive biomarkers of clinical response to lenvatinib therapy in hepatocellular carcinoma” reports a combination of increase of FGF19 and a decrease of Ang-2 in serum after a short period of treatment can provide prognosis for efficacy of lenvatinib therapy for HCC growth. The results from clinical samples demonstrate a convincing predictive value for lenvatinib therapy for HCC patients. Although there is a clear separation of progression-free survival with these measurements, whether overall survival can also be segregated from these measurements is not known. In addition, the significance of this study in providing valuable time for alternative and savage therapy for non-responders was not discussed. Neither was there mechanistic dissection of these clinical findings, particularly how to improve the efficacy and/or alternative therapy for lenvatinib non-responders. This study appears to be suitable for a more specialized clinical journal.
Author Response
We are grateful to the reviewers for the critical comments and useful suggestions that have helped us to improve our paper. As indicated in the responses that follow, we have taken all these comments and suggestions into account in the revised version of our paper.
We added Satoshi Kobayashi and Nobuhiro Hattori as a coauthor, and have revised the Conflicts of Interest statement as follows (Lines 463-465):
“Hidenori Toyoda has received lecture fees from AbbVie, MSD and Bayer Yakuhin Ltd. Atsushi Hiraoka has received lecture fees from Eisai Co., Ltd., Bayer Yakuhin Ltd., and Otsuka Pharmaceutical Co., Ltd.”
The manuscript by Chuma et al. “Early changes in circulating FGF19 and Ang-2 levels as possible predictive biomarkers of clinical response to lenvatinib therapy in hepatocellular carcinoma” reports a combination of increase of FGF19 and a decrease of Ang-2 in serum after a short period of treatment can provide prognosis for efficacy of lenvatinib therapy for HCC growth. The results from clinical samples demonstrate a convincing predictive value for lenvatinib therapy for HCC patients.
Although there is a clear separation of progression-free survival with these measurements, whether overall survival can also be segregated from these measurements is not known. In addition, the significance of this study in providing valuable time for alternative and savage therapy for non-responders was not discussed. Neither was there mechanistic dissection of these clinical findings, particularly how to improve the efficacy and/or alternative therapy for lenvatinib non-responders. This study appears to be suitable for a more specialized clinical journal.
Response
As suggested by the reviewer, we analyzed the association between serum FGF19 and Ang-2 levels and overall survival (OS). Kaplan-Meier analysis revealed a higher OS rate in patients with both an increase in FGF19 (FGF19-i) and a decrease in Ang-2 (Ang-2-d), as compared to those with either FGF19-i alone or Ang-2-d alone, and to those with neither FGF19-i nor Ang-2-d. However, OS still did not reach the median. During the study period, 26 patients developed progressive disease (PD). After developed PD, of the 26 patients, 15 patients received best supportive care, three patients received sorafenib treatment, one patient received regorafenib treatment, three patients received hepatic arterial infusion chemotherapy, and four patients were continued lenvatinib therapy. Since the number of patients received alternative therapy was small and the observation period was relatively brief, we could not comment on the effect of alternative therapy for non-responders on OS in this study. Based on these points and the reviewer’s suggestion, we have added the following sentences in the revised text:
“Secondary, observation period is relative brief (median, 168 days; range, 38–474 days). We performed Kaplan-Meier analysis and revealed that OS rate in patients with both FGF19-i and Ang-2-d (1-year OS, 85.7%) than in either FGF19-i alone or Ang-2-d alone patients (1-year OS, 62.2%), and in those with neither FGF19-i nor Ang-2-d (median OS 200 days P=0.0329; Supplementary Figure 3). Because of relative short periods of observation period, we could not mention whether these markers were involved in overall survival. Furthermore, we did not refer to alternative therapy for non-responders could provide valuable time. Further longer observation is needed to whether these markers are predictive OS and improve the efficacy by alternative therapy for lenvatinib non-responders.” (lines 334-342)
Supplementary Figure 3. Association between changes in FGF19 and Ang-2 levels and overall survival. Kaplan-Meier analysis of 74 hepatocellular carcinoma (HCC) patients who received lenvatinib treatment, stratified according to ratio of FGF19 and Ang-2. FGF19-i represents a FGF19 ratio at 4 weeks/baseline of >1.51 and Ang-2-d represents an Ang-2 ratio at 4 weeks/baseline of < 0.67.
Round 2
Reviewer 1 Report
The authors responded to my doubts and questions posed by myself following reading of the first version of the paper. All the data or corrections have been introduced everywhere where they were required. I hope that in this form the paper can be accepted for publication.
Reviewer 2 Report
The revision is improved for publication.